# Variations of Supercooling Capacity in Intertidal Gastropods

**DOI:** 10.3390/ani13040724

**Published:** 2023-02-17

**Authors:** Jie Wang, Shuo Wang

**Affiliations:** 1Key Laboratory of Mariculture, Ministry of Education, Fisheries College, Ocean University of China, Qingdao 266003, China; 2Function Laboratory for Marine Fisheries Science and Food Production Processes, Qingdao National Laboratory for Marine Science and Technology, Qingdao 266235, China

**Keywords:** cooling rate, low temperature, molluscs, species distribution, supercooling point

## Abstract

**Simple Summary:**

Intertidal snails can survive freezing (i.e., tolerate the formation of ice in their body) at very low subzero temperatures. The temperature at which body fluids spontaneously freeze (supercooling point, SCP) varies among intertidal snails, potentially depending on their geographic and vertical distributions. The objective of this study was to understand the variations of SCP in common intertidal snails in China. We compared nine intertidal species with different geographic and vertical distributions. Results indicated that at the intraspecies level, the higher the cooling rate of an individual was, the higher the SCP was. High-shore species living in Northern China showed lower SCPs, while southern high-shore species had higher SCPs. These results suggested that cooling rate and local winter temperatures contribute to the difference in SCPs within and between species and might be useful for evaluating and predicting species distribution.

**Abstract:**

Winter low-temperature confines species distribution. Intertidal gastropods are distributed from tropical to polar zones, facing variable intensities and durations of low temperatures. They usually set their supercooling points (SCPs) at high subzero temperatures to stimulate freezing. However, the variations in SCP in intertidal gastropods at intraspecific and interspecific levels remain poorly understood. Here, we measured the body size, cooling rate, and SCP of nine intertidal gastropod species in China. These species were distributed in high or middle intertidal zone with different geographic distributions. The average SCPs (−4.27~−7.10 °C) and the coefficients of variation of SCP (22.6%~45.9%) were high in all species. At the intraspecific level, the supercooling capacity was positively correlated with the cooling rate. Interspecifically, the supercooling capacity was closely related to the cooling rate, and also to the species’ geographical distribution. Northern high-shore species showed lower SCPs, while southern high-shore species had higher SCPs. There was no difference in SCP between widespread high- and mid-shore species. Our results indicated that the supercooling capability is potentially an adaptative response to the local winter temperatures, and the cooling rate is a factor in determining the difference in SCP at the intraspecific and interspecific levels.

## 1. Introduction

Low temperature in winter defines life at high latitudes and altitudes. The rocky intertidal ecosystem has the most physically harsh environments [1], but it is filled with different species with varied latitudinal and vertical distributions. Abiotic factors such as temperature mainly determine the upper limits [2,3], while ecological interactions such as predator–prey interaction usually set species’ lower limits [4,5,6]. In comparison to considerable work that has been completed to reveal the effects of high temperatures on species distribution [2,3,7,8], the ecological implications of low temperatures in winter remain inadequately understood [9,10]. Winter conditions have high ecological and physiological significance on intertidal communities [11,12]. For example, winter ecological processes exhibit crucial influences on reproduction, growth, survival, fitness, physiological performance, and distribution of intertidal species [11,13]. As interest grows in winter ecology and in the context of northward distribution shift, it is necessary to understand the strategy and ability of intertidal species to cope with low temperature and their effects on distribution at both local and geographic scales.

Intertidal species have to frequently contend with extremely low temperatures in winter, especially during the aerial exposure period [14]. When exposed to subzero temperatures, intertidal animals must survive the formation of ice [15] and have evolved different strategies according to the environmental characteristics of their habitats and their mobility [16,17,18]. Overall, there are two main strategies: one is freezing avoidance (i.e., organisms avoid freezing by extensive supercooling), and another is freezing tolerance (i.e., organisms survive freezing of the body fluids). Supercooling reflects an ability to avoid freezing injury. In freeze-avoiding ectotherms, the supercooling point (SCP) is usually lower than the lethal temperature (LLT). In the freezing-tolerant species, the capacity to supercool is limited and the SCP is higher than the LLT. Marine, freshwater, or terrestrial gastropods display different strategies like avoidance, partial tolerance, or full tolerance. Among intertidal animals, nearly all studied intertidal gastropods are freezing tolerant [17,19]. Strategic consistency in intertidal gastropods might reflect their adaptations to less predictable and highly variable intertidal environments.

Thermal tolerance of intertidal species varies significantly across different vertical and geographic distributions owing to the variations in intensity and duration of thermal stress [20,21]. In comparison to animals on the lower shore, high-shore organisms are more likely tolerant to extreme temperatures [22,23,24,25]. Therefore, high intertidal species are suggested to be better adapted to heat or freeze exposures. On a larger scale, higher latitude species or populations usually display increased cold tolerance [26,27,28,29] since they are frequently exposed to subzero temperatures in winter. Due to heterogeneous thermal regimes at both local and geographic scales, organisms often adapt their thermal tolerance in response to the local temperature [2,30,31]. However, the relationship between the capacity to supercool and/or cold tolerance and species distribution remains understudied [29,32,33,34].

Species’ cold tolerance is affected by multiple environmental factors, such as temperature, salinity, and oxygen tension [17,35,36]. When extracellular ice formation occurs during freezing, intracellular osmolytes (e.g., betaine, glycine, and alanine) can act as cryoprotection to prevent cellular dehydration and help cope with the osmotic shock due to freezing [37]. Since high salinity acclimation elevates the concentration of intracellular osmolytes, intertidal invertebrates can increase their freezing tolerance after acclimation to high salinity [23,38]. Acclimation to lower temperatures can also increase the capacity of cold tolerance of intertidal species [23,26]. In addition to the effects of intensity and duration of low temperatures [36], the cooling rate also affects freezing tolerance [16,39]. For example, larger specimens of *Littorina littorea* would be more tolerant to cold stress due to lower cooling rates [40]. There are obvious interspecific variations in body size among intertidal gastropods. Whether body size is an important factor determining supercooling and/or freezing tolerance of intertidal species at the interspecific and intraspecific levels remains poorly understood.

To investigate the relationship between the capacity of supercooling and various factors (including abiotic and biotic factors), we compared differences in SCPs among nine intertidal gastropod species with different geographic and vertical distributions along the Chinese coastline and hypothesized that (1) the high-latitude species should show lower SCPs; (2) the high shore species should show higher SCPs due to smaller body sizes and relatively high cooling rates. These findings might be helpful for understanding the cold tolerance of intertidal species and for predicting species’ biogeographic distribution under climate change.

## 2. Materials and Methods

### 2.1. Sample Collection

Nine intertidal gastropod species were randomly collected on the rocky shores in China in October 2021 (n = 26~49). The northern species and widespread species were collected in Qingdao (36.26° N, 120.69° E). Two southern species, high intertidal species *Echinolittorina malaccana* and mid-intertidal species *Nerita yoldii* were collected from Dongtou (27.86° N, 121.18° E) (more information in Figure 1a; Table 1). After collection, samples were transported back to the Key Laboratory of Mariculture, Ocean University of China within 48 h. Snails were cultured at a temperature of 16 °C for ~7 days under a tidal cycle with 6 h of emersion and 6 h of immersion. This acclimation temperature was used to simulate the average air temperature in October in Qingdao. This 7-day short-term acclimation was designed to eliminate variable thermal history of individuals [41].

### 2.2. Environmental Temperature Variations

To understand winter temperature variation in two sampling locations, we downloaded air temperature data from the National Center for Atmospheric Research (NCAR, https://rda.ucar.edu, accessed on 15 December 2021), and analyzed the air temperature of winter months (December, January, and February) between Qingdao and Dongtou from 2011 to 2021.

### 2.3. Measurement of Shell Size

Shell length (*L*), height (*H*), and width (*W*) (Figure 2a) were measured by using a digital Vernier caliper with 0.01 mm accuracy. Shell volume (*V*) was calculated according to the formula as previous studies described [46,47]:$$V=2\left[\frac{\pi }{3}\left(\frac{L}{2}\right)\left(\frac{W}{2}\right)\left(\frac{H}{2}\right)\right]$$

### 2.4. Freezing Curve Measurement

Gastropods were first wiped with paper towels to dry seawater. The tip of a sensor cable (0.85 mm in diameter) from the multichannel temperature recorder (YP5016G, Shenzhen YongPeng Instrument Corporation, Shenzhen, China) was gently slipped into its shell once the animal extended its foot. This recorder can continuously log temperature at intervals of one second with a resolution of ±0.1 °C. Then, the individual was placed upright in an aluminum radiant plate at −14 °C, a temperature that represented an extremely low air temperature that gastropods may encounter in winter in Qingdao. The recorder logged temperature changes from 16 °C to −14 °C.

A typical freezing curve of an individual was shown in Figure 3. The supercooling point (SCP) was the temperature at which the bulk of ice formation occurred and generated the exothermic release of heat as the tissue water was converted to ice [48,49]. *T* was the temperature at which an individual reached its supercooling point and *t* was the time when ice formation occurred in the body. The cooling rate was calculated according to *T* and *t* values.

### 2.5. Statistical Analysis

Due to the violation of variance homogeneity (Bartlett test) and/or normality (Shapiro–Wilk test), the nonparametric Kruskal–Wallis test was performed to determine significant differences in volume, cooling rate, and SCP among species. Wilcoxon rank sum test was used to make pairwise comparisons with Bonferroni adjustment. Within each species, linear regression was used to determine the relationship between cooling rate and volume, and between SCP and cooling rate. A *p*-value lower than 0.05 was considered significant. All analyses and visualizations were performed in R v4.0.2 [50].

Generalized additive models (GAMs) were applied to analyze the relationship between SCP values and five explanatory variables (geographic distribution, vertical distribution, species, body size, and cooling rate). Continuous predictors, which were regarded as smooth terms, applied the default degrees of freedom in the fitting procedure. The five explanatory variables were added to the GAM model one by one to observe the changes in model parameters (AIC value, residual deviance, and cumulative explained deviance). The final GAM model could be written as an Equation: SCP~factor(geographic distribution) + factor(vertical distribution) + factor(species) + s(body size) + s(cooling rate).

## 3. Results

### 3.1. Air Temperatures in Winter

The air temperature data in winter from 2011 to 2021 showed that the intensity and frequency of cold temperatures (<0 °C) in Qingdao were much higher than that in the southern location, Dongtou (Figure 1b). During 2011–2021, the lowest temperature reached −14.3 °C in Qingdao, while it was −1.7 °C in Dongtou. The location Qingdao was subject to a high frequency of temperature below −4 °C in winter, a temperature representing the average of SCPs (−4.3 °C, see Section 3.4) in studied species.

### 3.2. Body Size of Intertidal Snails

The body size (i.e., the volume here) was significantly different among intertidal gastropods (Kruskal–Wallis test, *df* = 8, χ^2^ = 228.54, *p* < 2.2 × 10^−16^) (Figure 2b). For the southern and widespread species in China, the average volume of body size of high-zone species (0.05~0.07 cm^3^) was at least ten times smaller than that of species in the mid-intertidal zone (0.58~1.41 cm^3^) (*p* < 0.05). In Northern China, the high-zone species (mean ± SD: 0.67 ± 0.11 cm^3^) was nearly one and a half times smaller than the mid-intertidal species (1.10 ± 0.22 cm^3^) (*p* < 0.05).

### 3.3. Cooling Rate Variations

The cooling rate was significantly different among species (Kruskal–Wallis test, *df* = 8, χ^2^ = 205.5, *p* < 2.2 × 10^−16^), with high-zone species showing a higher cooling rate (Figure 4). Among the southern and widespread species, the cooling rate in the high-zone species (mean ± SD: *E. malaccana*, 34.89 ± 7.83 °C/min; *E. radiata*, 35.92 ± 14.43 °C/min; *Littoraria sinensis*, 28.26 ± 11.49 °C/min) was significantly higher than that in the mid-intertidal species (*N. yoldii*, 8.11 ± 2.59 °C/min; *Reishia clavigera*, 7.13 ± 1.85 °C/min; *Monodonta labio*, 5.29 ± 1.29 °C/min; *Tegula rustica*, 4.99 ± 1.38 °C/min) (*p* < 0.05). For the northern species, there was no significant difference between high-zone (*Littorina brevicula*, 7.27 ± 1.69 °C/min) and mid-intertidal (*Lunella coreensis*, 6.10 ± 1.31 °C/min) species (*p* = 0.38).

The cooling rate decreased with increasing body size (Figure 5). The cooling rate significantly reduced with the increase in body size in all high-zone species (*E. radiata*, *F* = 22.64, *p* < 0.05; *L. sinensis*, *F* = 12.20, *p* < 0.05; *L. brevicula*, *F* = 5.58, *p* < 0.05; *E. malaccana*, *F* = 11.16, *p* < 0.05). Three high-zone species with the lowest average body size (*E. malaccana*, *E. radiata*, and *L. sinensis*) had a decreasing rate ranging from −0.16 to −0.58 °C/mm^3^, which was over ten times faster than other species (<−0.01 °C/mm^3^). Two out of five mid-intertidal species displayed a significant decreasing rate (*T. rustica*, *F* = 35.07, *p* < 0.05; *N. yoldii*, *F* = 7.95, *p* < 0.05), reaching −0.003~−0.01 °C/mm^3^.

### 3.4. Supercooling Point Variations

There was variation in SCP among species (Kruskal–Wallis test, *df* = 8, χ^2^ = 51.89, *p* = 1.77 × 10^−8^) (Figure 6). In Southern China, the high-zone species *E. malaccana* (mean ± SD: −4.27 ± 0.99 °C) had significantly higher SCP in comparison to the mid-intertidal *N. yoldii* (−7.10 ± 1.87 °C) (*p* < 0.05). In Northern China, the high-zone *L. brevicular* (−6.60 ± 1.54 °C) presented significantly lower SCP than the mid-intertidal *L. coreensis* (−4.78 ± 1.08 °C) (*p* < 0.05). There was no significant difference in SCP value among the widely distributed species (−5.74 ± 2.01~−6.33 ± 2.41 °C, *p* > 0.05).

In comparison to the coefficient of variation (CV) of SCP among species (14.1%), there were high variations in SCP within species. The lowest CV was in *L. coreeniss* (22.6%), and the largest CV occurred in *E. radiata* (45.9%). At the intraspecies level, the SCP increased as the cooling rate elevated in most species (Figure 7).

### 3.5. Factors Contributing to the Supercooling Points

GAM modeling results showed that four of five variables (including geographic distribution, vertical distribution, species, and cooling rate) were identified as significant (Table 2). The six variables contributed 39.00% of the total variability in the deviance of SCP. Among these variables, the cooling rate was the main factor in affecting SCP, accounting for 23.10% of the total variability in deviance (*edf* = 3.95, *F* = 9, *p* < 0.01). Species, geographic distribution, and vertical distribution contributed 11.22% (*df* = 8, *F* = 54, *p* < 0.01), 4.74% (*df* = 2, *F* = 145, *p* < 0.01), and 0.94% (*df* = 1, *F* = 16, *p* < 0.01) of the variability of SCPs, respectively.

## 4. Discussion

Understanding the ability to supercool intertidal species and its influencing factors is crucial for evaluating and predicting species distribution change in the context of climate change. In the present study, intertidal gastropods showed a low ability to supercool, which is mainly affected by individual cooling rates within species. Geographical location and tidal height can also partially explain the variations in SCPs among species.

### 4.1. Limited Capacity to Supercool of Intertidal Gastropods

Intertidal gastropods frequently suffer from freezing temperatures in winter. In the present study, the SCPs of intertidal gastropods were −4.27~−7.10 °C. These results are consistent with previous results that intertidal species usually have moderate to high SCPs [51], such as the snail *L. brevicula* (−7.96 °C) [26] and *Nucella lamellosa* (−3.99 °C) [52], the bivalve *Mytilus trossulus* (−5.50 °C) [23], except for the limpet *Patinigera polaris* (−10 °C) [53].

The limited supercooling capability cannot meet the challenge of low temperatures in winter in Northern China. As we mentioned above, intertidal species in Qingdao frequently suffer from extremely low temperatures below the SCPs, implying that most intertidal gastropods are frozen frequently in winter and adopt a freezing tolerance strategy for survival. Freezing can cause tissue, cellular, and protein damage [16,37,54], and ice formation at moderate to high subzero temperatures is not beneficial for the survival of organisms. Nearly all other reported intertidal species also adopt a freezing tolerance strategy (see a review in [17]), with one exception in an Antarctic limpet *P. Polaris* using a freezing avoidance strategy [53]. Compared with intertidal gastropod, terrestrial gastropod adopts partial tolerance or avoidance strategy [17]. Although it is difficult for intertidal species to avoid freezing in winter [55], most intertidal organisms would not be exposed to extremely low air temperatures for a long period of time due to the tidal cycle. Therefore, two strategies (freezing avoidance and freezing tolerance) should be adopted together to deal with freezing in winter [17,36].

### 4.2. Variation in SCP at the Geographic and Local Scales

At the geographic scale, high-latitude species did not show expected lower SCPs compared to low-latitude species. The species living in Northern China had similar capacity of supercooling in spite of significant differences in body size and cooling rate among them, implying that some species and/or individuals in high latitudes could reduce their SCPs to resist ice formation. For the high-latitude intertidal species and/or populations, since they are frequently exposed to extreme subzero temperatures [14], they prefer to reduce their SCPs [56] and be more freezing tolerant [17,26].

As for the low-latitude species, it seems unnecessary for southern species to evolve a high capacity to resist subzero temperatures due to the benign thermal environment. Theoretically, southern species would not positively reduce their SCPs and thereof their capacities to supercool are mainly controlled by the body size. High SCPs were indeed observed in the southern species *E. malaccana*, which has a small body size and rapid cooling rate. Interestingly, another southern species *N. yoldii* showed low SCPs like the northern and widespread species. This might be linked to its large body size and low cooling rate. Such a high capacity to supercool in *N. yoldii* might be one of the reasons that cause its northward expansion in China [34,57].

The variations in SCP among high-shore species might be associated with the adaptation to local thermal conditions in winter, such as the intensity and duration of subzero temperatures. Although the high-shore species *L. brevicular* in Qingdao was smaller in size, it displayed significantly lower SCP than the mid-shore species *L. coreensis*. Since high-shore species would suffer from higher frequency and longer exposure to subzero temperatures, increased capacity to supercool would help them reduce the potential negative effects of ice formation as described in some intertidal gastropods inhabiting polar regions. For example, the SCP significantly decreased from low-shore *N. lamellosa*, and mid-shore *N. lima*, to high-shore *L. Sitkana* in Alaska [58].

In the present study, we also noticed that there was no significant difference in SCP among three high-shore species in Qingdao, despite their significant difference in size and cooling rate. Two widespread high-shore species (*E. radiata* and *L*. *sinensis*) in Qingdao might have evolved robust capacity or related mechanisms to lower their SCP to resist freezing like the northern high-shore species (*L. brevicular*). Historically, following the glacial retreat, demographic expansion from low to high latitudes has been observed in many marine taxa [59]. Northward expanding population would gradually evolve an ability to resist subzero temperatures (such as reducing its SCP), and therefore poleward populations usually show higher SCP and/or freeze tolerance than their equatorward populations (e.g., barnacle *Semibalanus balanoides*) [56]. This speculation could be tested by making a comparison between northern and southern populations of widespread species in a future study.

### 4.3. Cooling Rate Leads to Variation in SCPs

The cooling rate can affect the ability to supercool intertidal gastropods. Cooling rate, which is controlled by body size and exposure temperature, has been suggested as a critical factor in determining species’ freezing tolerance [36]. The cooling rate can affect the duration of the cooling process, in which organisms would synthesize or recruit protecting agents (e.g., cryoprotective compounds and ice-binding proteins) to reduce SCP and resist freezing stress [60,61] and therefore influence the ability of freeze tolerance.

Variations in physiological adaptations among species might explain the difference in the cooling rate and the capacity of supercooling [62,63]. For example, tissue components, such as muscle fibers and high-molecular-weight glycoproteins, could improve species’ resistance to ice formation. Increasing the production of factors that inhibit lipid peroxidation can also resist the formation of extracellular ice. Detecting physiological mechanisms leading to the divergence of the ability of supercooling is needed in our future studies. One common biochemical mechanism for low-temperature tolerance is the presence of ice-binding protein (IBP), which can act as an antifreeze [60]. Since IBPs might evolve through gene duplication in invertebrates in response to freezing risk [64], future work can focus on the expression and evolution of IBPs in affecting the capacity to supercool to explain the variation in SCP across intertidal species.

## 5. Conclusions

This study demonstrated that there is a high variation in supercooling capacity in intertidal gastropods at intraspecific and interspecific levels in China. Cooling rate and local winter temperatures are the main factors leading to the difference in SCPs within and between species. However, this finding needs to be interpreted with caution since the present study only analyzed one sampling occasion in October and a short-term acclimation (7 days) might not completely eliminate the variable environment history of individuals. Despite these limitations, our findings could be useful for predicting future distributions of intertidal species under global warming. Future studies may collect samples seasonally and investigate their changes in body size, cooling rate, and SCPs over time, which provides insight into temporal and spatial patterns of variation in supercooling capacity and underlying mechanisms causing the difference.

## Figures and Tables

**Figure 1 animals-13-00724-f001:**
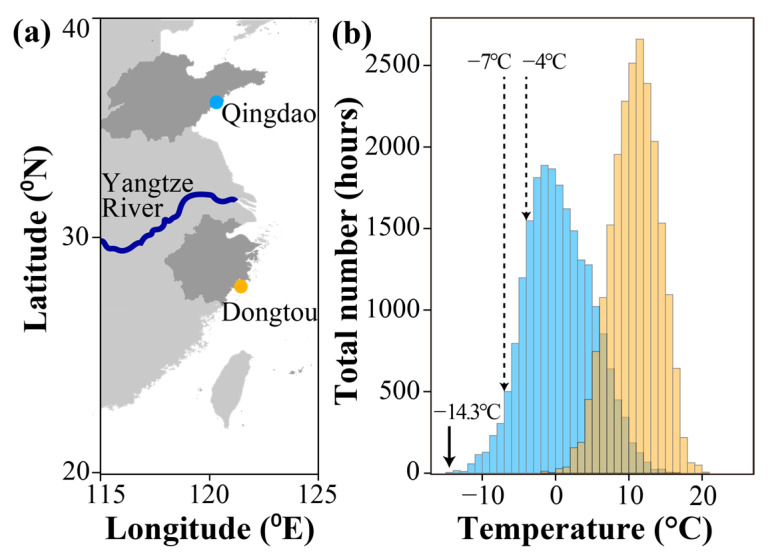
Sampling locations in China and winter temperatures at sampling locations. (**a**) Two sampling locations (Qingdao and Dongtou) of nine intertidal gastropod species. (**b**) The total number of occurrences of temperatures in winter (December, January, and February) over the past ten years (2011–2021) in Qingdao (blue bars) and Dongtou (orange bars).

**Figure 2 animals-13-00724-f002:**
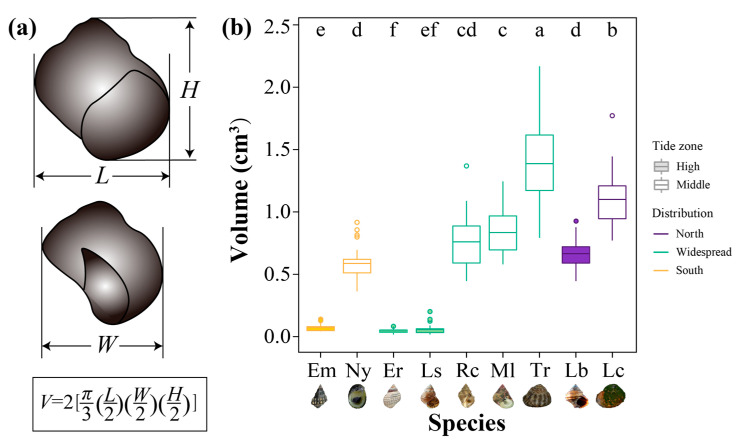
Body size measurement in snails and the comparison of body size among nine species. (**a**) Shell diagrams indicate the measurement of length (*L*), width (*W*), and height (*H*). Volume (*V*) is calculated according to the formula below. (**b**) Volumes of nine intertidal gastropods with different geographic distribution ranges and tide zone (*Echinolittorina malaccana*, Em; *Nerita yoldii*, Ny; *E. radiata*, Er; *Littoraria sinensis*, Ls; *Reishia clavigera*, Rc; *Monodonta labio*, Ml; *Tegula rustica*, Tr; *Littorina brevicula*, Lb; *Lunella coreensis*, Lc). Box plots show median (center line within the box), first and third quartile values (lower and upper borders of the box), and the whiskers extend to 1.5× the interquartile range. The spots outside the box are outliers. Different letters indicate significant differences in volume among species (*p* < 0.05).

**Figure 3 animals-13-00724-f003:**
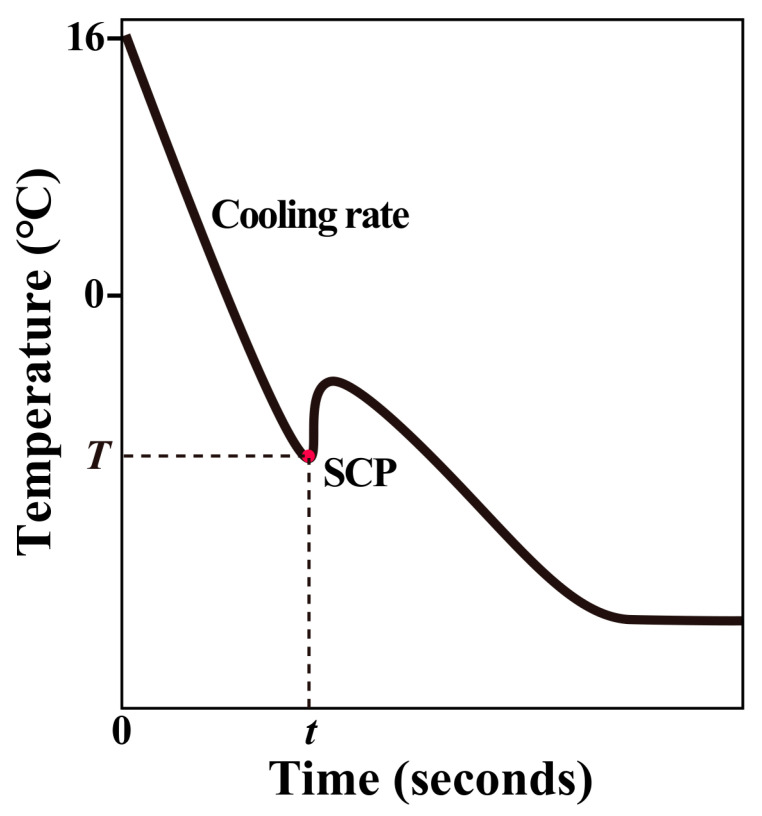
Example of observed changes in body temperature of a snail that was cooled from 16 °C. The supercooling point (SCP) denotes the temperature at which the snail froze. *T* is the temperature at which an individual reaches its supercooling point and *t* is the time when ice formation occurs in the body. Cooling rate is calculated based on *T* and *t*.

**Figure 4 animals-13-00724-f004:**
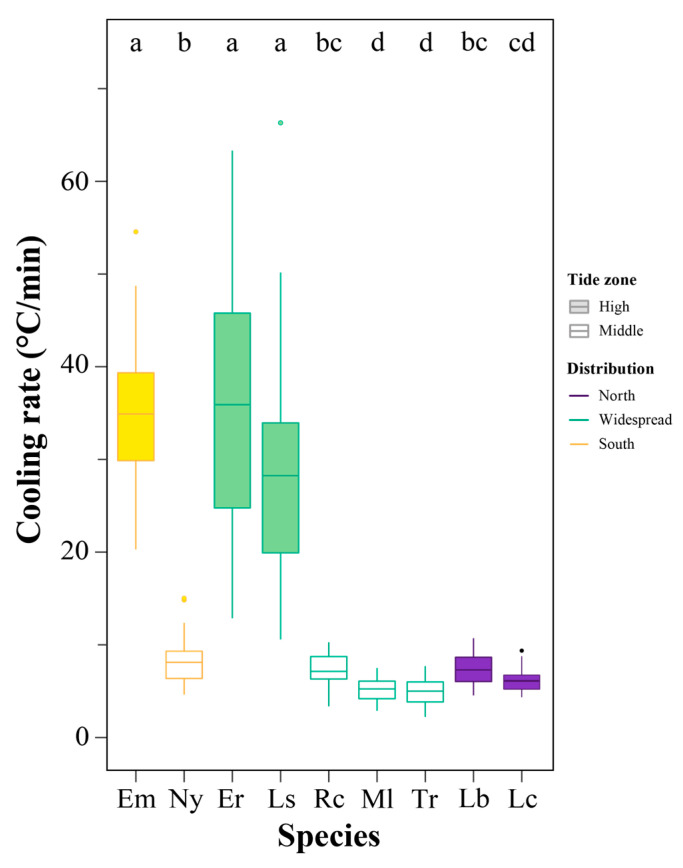
Cooling rates of nine intertidal species in the present study. Different letters indicate significant differences in cooling rate among species (*p* < 0.05). Species abbreviations: *Echinolittorina malaccana*, Em; *Nerita yoldii*, Ny; *E. radiata*, Er; *Littoraria sinensis*, Ls; *Reishia clavigera*, Rc; *Monodonta labio*, Ml; *Tegula rustica*, Tr; *Littorina brevicula*, Lb; *Lunella coreensis*, Lc.

**Figure 5 animals-13-00724-f005:**
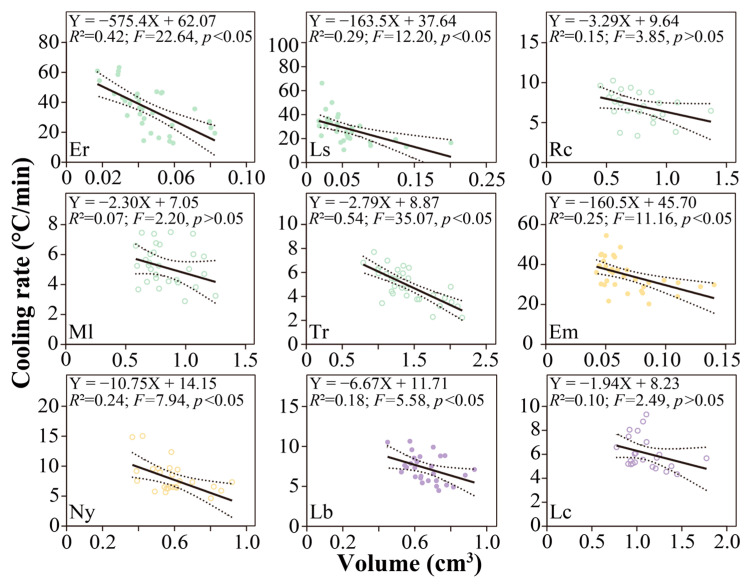
Correlation between individual’s volume and cooling rate in each species. Northern, southern, and widespread species are distinguished with purple, orange, and green, respectively. Filled circles and empty circles represent high-shore individuals and mid-shore individuals, respectively. *R*-squared reflects the goodness of fit. The *F* and *p* values indicate whether the slope is significantly non-zero. Solid line reflects linear regression line, and dashed lines reflect the 95% confidence intervals around linear regression line. Species abbreviations: *Echinolittorina malaccana*, Em; *Nerita yoldii*, Ny; *E. radiata*, Er; *Littoraria sinensis*, Ls; *Reishia clavigera*, Rc; *Monodonta labio*, Ml; *Tegula rustica*, Tr; *Littorina brevicula*, Lb; *Lunella coreensis*, Lc.

**Figure 6 animals-13-00724-f006:**
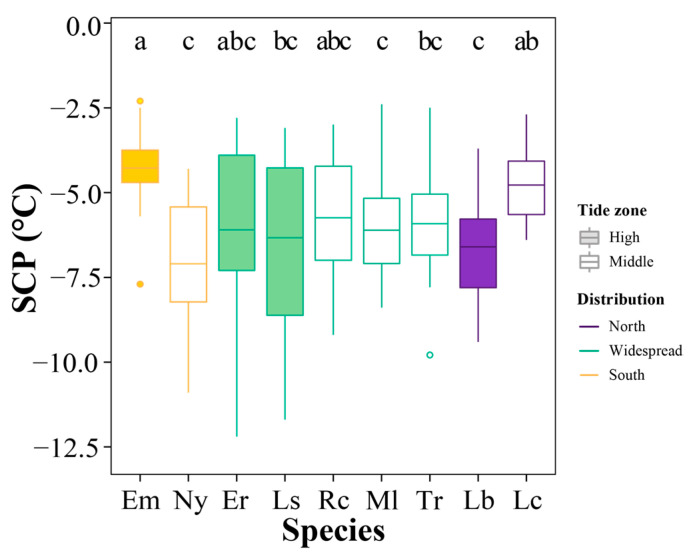
Supercooling points (SCPs) of nine intertidal species in China. The center line in the box plot is the median value, and the whiskers extend to 1.5× the interquartile range (IQR). The spots outside the box are outliers. Different letters indicate significant differences in cooling rate among species (*p* < 0.05). Species abbreviations: *Echinolittorina malaccana*, Em; *Nerita yoldii*, Ny; *E. radiata*, Er; *Littoraria sinensis*, Ls; *Reishia clavigera*, Rc; *Monodonta labio*, Ml; *Tegula rustica*, Tr; *Littorina brevicula*, Lb; *Lunella coreensis*, Lc.

**Figure 7 animals-13-00724-f007:**
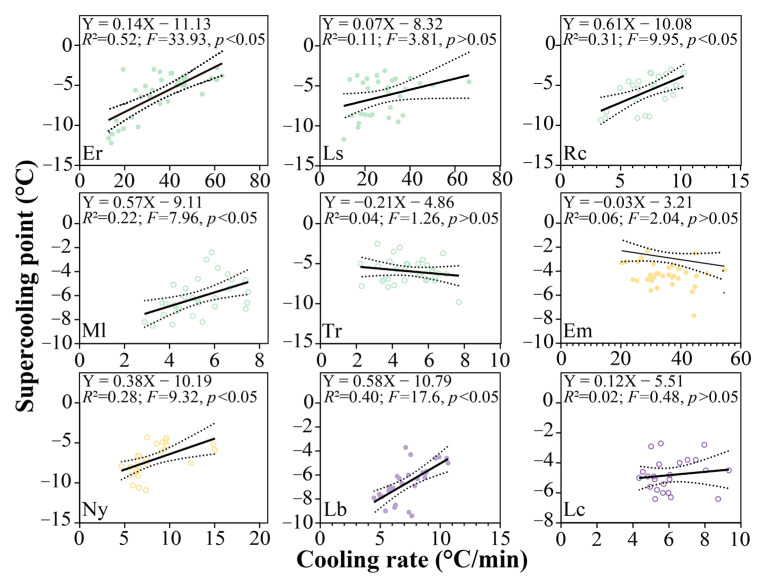
Relationship between cooling rate and supercooling point in each study species. The nine gastropods are northern (purple), southern (orange) and widespread (green) species, respectively, and live in high-shore (filled circle) or mid-shore (empty circle) zones. *R*-squared reflects the goodness of fit. The *F* and *p* values indicate whether the slope is significantly non-zero. Solid line reflects linear regression line, and dashed lines reflect the 95% confidence intervals. Species abbreviations: *Echinolittorina malaccana*, Em; *Nerita yoldii*, Ny; *E. radiata*, Er; *Littoraria sinensis*, Ls; *Reishia clavigera*, Rc; *Monodonta labio*, Ml; *Tegula rustica*, Tr; *Littorina brevicula*, Lb; *Lunella coreensis*, Lc.

**Table 1 animals-13-00724-t001:** Information on the geographic and vertical distribution of nine intertidal molluscs in China.

Species	Geographic Distribution		Vertical Distribution (Height above CD) ^¶^		References
*Echinolittorina malaccana*	20° S~30° N	Southern species	3.00 m (1.75~3.25 m)	High-shore species	[42]
*E. radiata*	12° N~43° N	Widespread species	2.75 m (1.75~3.25 m)	High-shore species	[42]
*Littoraria sinensis*	19° N~40° N	Widespread species	Between *E. radiata* and *L. brevicula*	High-shore species	Present study
*Littorina brevicula*	22° N~45° N	Northern species	2.30 m (1.00~2.60 m)	High-shore species	[43]
*Nerita yoldii*	21° N~33° N	Southern species	1.75 m (1.00~2.50 m)	Mid-shore species	[44]
*Reishia clavigera*	6° S~40° N	Widespread species	1.00 m (1.00~2.25 m)	Mid-shore species	[44]
*Monodonta labio*	25° S~42° N	Widespread species	1.50 m (1.00~2.00 m)	Mid-shore species	[44]
*Tegula rustica*	17° S~44° N	Widespread species	Lower than *M. labio*	Mid-shore species	[45]
*Lunella coreensis*	30° N~42° N	Northern species	1.50 m (1.00~1.75 m)	Mid-shore species	[44]

^¶^: Main height of species distribution, in the parentheses were the range the species occurs.

**Table 2 animals-13-00724-t002:** Generalized additive model (GAM) analysis for supercooling point (SCP).

Added Term	*rdf*	*rd*	*df*	*edf*	*de*	*cde*	AIC
Null	283	1177.16	/	/	/	/	/
+Geographic distribution	281	1128.03	2	/	4.17%	4.17%	1205
+Vertical distribution	280	1102.44	1	/	2.18%	6.35%	1201
+Species	275	981.39	8	/	10.25%	16.60%	1178
+Volume	268	947.42	/	0.92	−0.70%	15.90%	1164
+Cooling rate	247	644.49	/	3.99	23.10%	39.00%	1024

*rdf*: residual degree of freedom; *rd*: residual deviance; *df*: degree of freedom; *edf*: effective degrees of freedom; *de*: deviance explained; *cde*: cumulation of deviance explained; AIC: Akaike information criterion.

## Data Availability

The data sets in the current study are available from the corresponding author upon reasonable request.

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
