# Peer review of "Variations of Supercooling Capacity in Intertidal Gastropods"

_animals, 2023, doi:10.3390/ani13040724_

Round 1

Reviewer 1 Report

I found this study quite interesting even if extremely focused on the physiology of gastropods, without functional ecology evaluations. The main limitation of the study is that despite it being experimentally well conducted, its value is affected by the sampling design that provides just one sampling in October, without the possibility of a seasonal evaluation of the starting biometric parameters of the animals. 

Keywords: Please try to avoid the use of words already reported in the Title. Try to substitute with some related ones.

The introduction section result rather wordy in some parts and is greatly focused on abiotic factors, while in my opinion, the distribution in the intertidal zones of gastropods and other related groups is also driven by ecological interactions among organisms. Even if this is not the topic of the manuscript, it should be considered a little in this part of the text. See for example:

10.1016/j.ecolmodel.2020.109206

10.4236/ojms.2016.62018 

Line 94: gastropod species

Figure 6: please add a legend to explain the species abbreviations.

Table 2: I suggest improving the clarity, reducing the size of the character to put them in one line of the text.

Lines 266-269: please, better argue this period exposing your results in comparison with references.

Line 308: double-check the style.

4.3 "Factors determining the supercooling ability": in the present form, this paragraph does not represent a Discussion, but an Introduction. It's impossible to relate these sections to the author's results, so it makes no sense.

The conclusion section needs to be rewritten based on the present study findings. The present form is so synthetic and vague, please link it more to your key findings in this study, and provide its limitations and possible future prospects.

Reviewer 2 Report

The paper presents an experimental analysis of supercooling capacity of nine species of intertidal gastropods. The manuscript is compact. The experimental work itself well organized and accompanied by adequate statistical analysis which permits to separate row of factors in GAM model: body volume, species, geographic site and intertidal position. Every such factor can contribute to supercooling capability.

 Experiments show interesting results. I’d like to clear up one point. The fact is that the results of physiological experiments are completely dependent on manipulations with the source material. Preparation for the cold season in living creatures is associated with a long period of physiological acclimation triggered by various factors, including the temperature regime. It is likely that molluscs from north location were already pre-acclimated in October (time of collection), in contrast to molluscs from a south population. The article provides data on winter temperatures, but does not indicate the local temperatures during the collection period. Is a weekly standard acclimation in the laboratory enough to " alignment" the physiological state of the molluscs from two sample sites? Probably, in the discussion section or in the material and methods, it would be worth adding explanations on this matter.

Otherwise, after correcting typos (in particular, in the names of species), the article can be recommended for publication.

Round 2

Reviewer 1 Report

Dear Authors,

I appreciated your care in revising this manuscript, better arguing some focal points and highlighting its limitations due to the experimental design. 

Best regards

The Reviewer